# A comparative transcriptomic analysis of replicating and dormant liver stages of the relapsing malaria parasite *Plasmodium cynomolgi*

Annemarie Voorberg-van der Wel[1†], Guglielmo Roma[2†], Devendra Kumar Gupta[3], Sven Schuierer[2], Florian Nigsch[2], Walter Carbone[2], Anne-Marie Zeeman[1], Boon Heng Lee[3], Sam O Hofman[1], Bart W Faber[1], Judith Knehr[2], Erica Pasini[1], Bernd Kinzel[2], Pablo Bifani[3], Ghislain M C Bonamy[3], Tewis Bouwmeester[2], Clemens H M Kocken[1]*, Thierry Tidiane Diagana[3]*

[1]Department of Parasitology, Biomedical Primate Research Centre, Rijswijk, Netherlands; [2]Novartis Institutes for BioMedical Research, Basel, Switzerland; [3]Novartis Institute for Tropical Diseases, Singapore, Singapore

*For correspondence:
kocken@bprc.nl (CHMK);
thierry.diagana@novartis.com
(TTD)

†These authors contributed equally to this work

**Abstract** *Plasmodium* liver hypnozoites, which cause disease relapse, are widely considered to be the last barrier towards malaria eradication. The biology of this quiescent form of the parasite is poorly understood which hinders drug discovery. We report a comparative transcriptomic dataset of replicating liver schizonts and dormant hypnozoites of the relapsing parasite *Plasmodium cynomolgi*. Hypnozoites express only 34% of *Plasmodium* physiological pathways, while 91% are expressed in replicating schizonts. Few known malaria drug targets are expressed in quiescent parasites, but pathways involved in microbial dormancy, maintenance of genome integrity and ATP homeostasis were robustly expressed. Several transcripts encoding heavy metal transporters were expressed in hypnozoites and the copper chelator neocuproine was cidal to all liver stage parasites. This transcriptomic dataset is a valuable resource for the discovery of vaccines and effective treatments to combat vivax malaria.
DOI: https://doi.org/10.7554/eLife.29605.001

## Introduction

*Plasmodium vivax* (*P. vivax*) is the major cause of malaria outside of Africa with an estimated 13.8 million malaria cases globally in 2015 (*World Health Organization (WHO), 2015*). Among *P. vivax* parasites' most salient biological features are the persisting dormant liver stages (hypnozoites) that can cause relapse infections and compromise future eradication programs (*Campo et al., 2015*). Although in vitro hepatic cultures systems for hypnozoite-forming parasites have been developed (*March et al., 2013*; *Zeeman et al., 2014*) and rodent models of humanized liver stage infections constituted recent advances (*Mikolajczak et al., 2015*), the search for new drugs targeting hypnozoites is hampered by our limited knowledge of this enigmatic dormant stage.

Microbes commonly employ cellular quiescence to survive environmental stresses such as starvation, immune surveillance, or chemotherapeutic interventions and for disease causing microbes, dormancy often underlies chronic infections that considerably complicate the clinical management of infected patients (*Rittershaus et al., 2013*). Cellular quiescence generally requires a physiological response underscored by a global repression of cellular metabolism but the preservation of mitochondrial respiration for ATP homeostasis and the maintenance of genome integrity (*Rittershaus et al., 2013*). Therapeutic interventions targeting some of these mechanisms have been

proposed for a limited number of human pathogens (*Andries et al., 2005*; *Rao et al., 2008*) but it is not clear whether *P. vivax* hypnozoites rely on similar physiological responses to survive in hepatocytes.

Some of the new drug targets that have been identified in the past decade (*McNamara and Winzeler, 2011*) have been shown to be critical in multiple stages of the parasite life cycle, such as PI4K (*McNamara et al., 2013*), DHODH (*Phillips et al., 2015*), eEF2 (*Baragaña et al., 2015*), and pheT-RNA (*Kato et al., 2016*). However, none has yet been shown to be a valid target for malaria radical cure and elimination of the hypnozoite in vivo. Little is known about the expression pattern of these drug targets during *Plasmodium* life cycle in the liver and more specifically, it is not clear whether these genes are expressed at all in dormant parasites.

Transcriptomics approaches to assess genome-wide gene expression levels of *Plasmodium* liver stage parasites are inherently challenging given the low infection grade ratios and the higher abundance of host cell transcripts. While previous reports have emerged providing a first glance of gene expression in *Plasmodium* liver stages (*Cubi et al., 2017*; *Vaughan et al., 2009*), we provide here a comprehensive dataset derived from green fluorescent protein (GFP)-tagged *Plasmodium cynomolgi* (*P. cynomolgi*) (*Voorberg-van der Wel et al., 2013*) — the nonhuman primate sister taxon of *P. vivax*, known to form hypnozoites (*Dembélé et al., 2014*; *Krotoski et al., 1982*). We have collected samples from multiple independent in vitro hepatocyte infections, containing thousands of purified hypnozoites and liver schizonts for RNA-Seq. The sequenced reads were mapped on the new high quality, completely annotated *P. cynomolgi* genome covering 7178 genes (*Pasini et al., 2017*). Using different approaches, we provide some preliminary validation of our comparative analysis of the transcriptome of replicating and quiescent liver-stages parasites that will constitute a valuable resource for the development of *P. vivax* vaccines and therapeutics.

## Results

### Hypnozoites express a smaller set of genes than schizonts

Six to seven days after *P. cynomolgi* sporozoite infection of primary simian hepatocytes, we FACS-purified hepatocytes containing hypnozoites and liver schizonts and prepared RNA for high-throughput sequencing (*Figure 1*; *Supplementary file 1*). After quality control, we excluded three samples due to their low number of parasite reads, resulting in a dataset containing three independent schizont samples and four independent hypnozoite samples for analyses (*Supplementary file 1*). To quantify parasite-specific expression for each *P. cynomolgi* gene, we determined the number of sequencing reads aligned to genes and computed gene expression values as the number of Fragments Per Kilobase per Million fragments mapped (FPKM) (*Schuierer and Roma, 2016*). *Supplementary file 2*, *3* respectively provide the list of reference genomes used and the analysis statistics. Overall, the raw gene expression values of the schizont samples are ~14 fold higher than those of the hypnozoite samples (*p*-value 1.1e-3). This global difference in gene expression between multi-nucleated schizonts and uni-nuclear hypnozoites could be partly attributed to differences in the number of parasite transcriptionally active units per hepatocyte, however it is not possible to determine this exact number. In order to account for this difference, we normalized the gene expression values against the total number of host reads per sample which we posit to represent a constant host RNA content across all samples (see Materials and methods; *Supplementary file 4*). All data reported in *Figures 1–4* show FPKM values after such normalization. A threshold of FPKM greater than one is deemed equivalent to one transcript copy per cell (*Mortazavi et al., 2008*). Using this threshold, hypnozoites generally express a lower number of genes compared to schizonts (respectively, 3308 vs 5702 genes at average FPKM per group $\geq$1). In addition, the expression level of these genes in schizonts is higher than in hypnozoites (average expression 89.14 and 9.88 FPKM, respectively) (*Figure 1C*). To further validate this finding, we carried out RNA fluorescence in situ hybridization (RNA-FISH) to quantitatively evaluate the expression of abundantly expressed genes at the single-cell level in liver stage cultures. In agreement with the RNA-Seq results, the RNA-FISH staining with probes for *gapdh* and *hsp70* showed a markedly lower level in hypnozoites compared to schizonts' expression (*Figure 1D*).

We then compared the gene expression data with those recently published by Cubi *et al.* (*Cubi et al., 2017*). Since the two studies used different reference genomes and annotation files, we

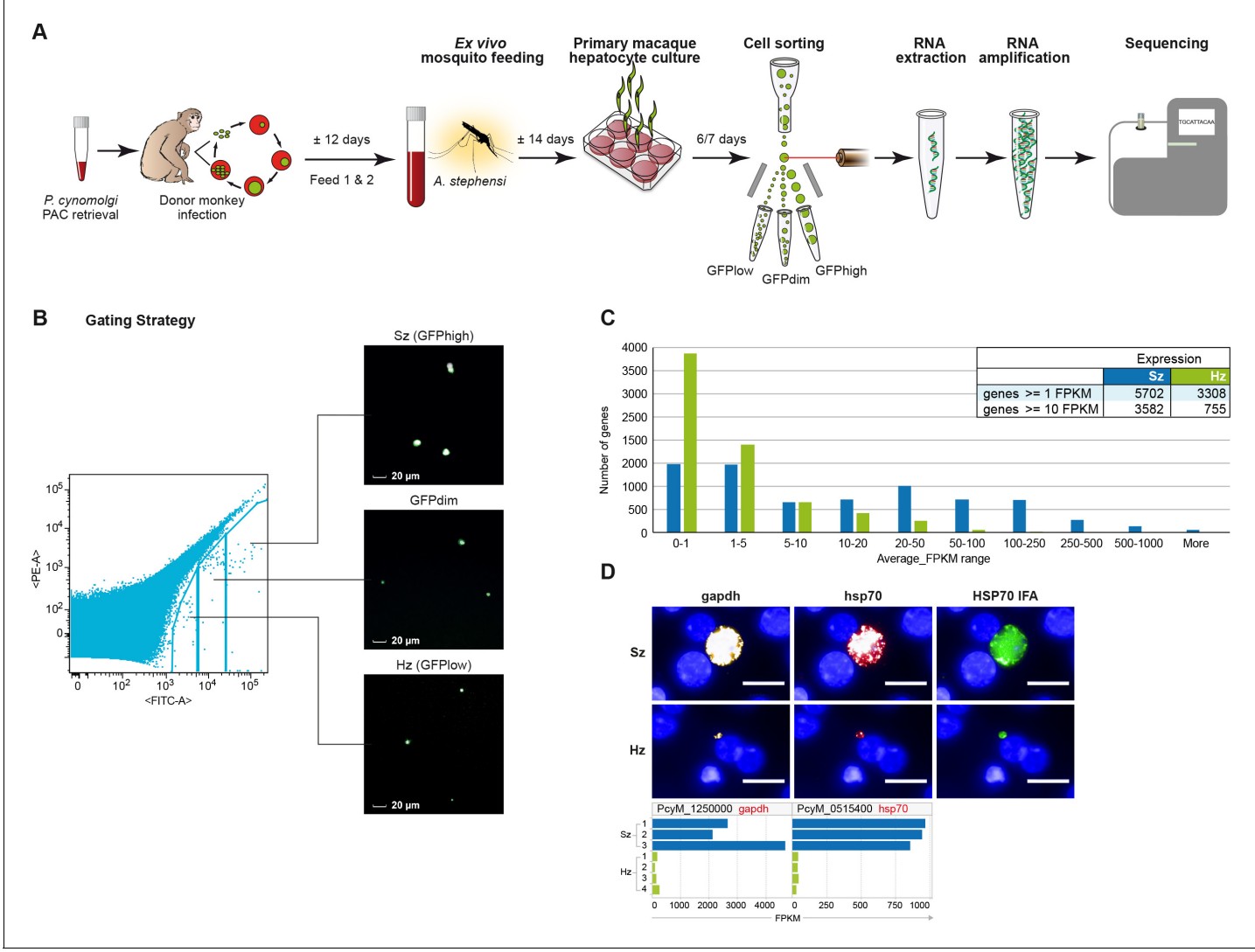

**Figure 1.** Transcriptomics of relapsing malaria liver stage parasites. (**A**) Scheme of experimental approach for purification and RNA-sequencing of cultured *P. cynomolgi* M malaria liver stage schizonts (Sz) and hypnozoites (Hz). To enable FACS purification, *P. cynomolgi* parasites that stably express GFP using a Plasmodium Artificial Chromosome (PAC) were used. For further details, see Materials and methods. (**B**) Gating strategy included an extra gate, 'GFPdim', not used in subsequent RNA-seq analysis to ensure a strict separation of 'GFPlow' and 'GFPhigh' parasites. (**C**) Distribution of average gene expression values in the hypnozoite (green; n = 4) and schizont (blue; n = 3) samples. FPKM, Fragments per kilobase of transcript per million mapped reads. (**D**) Top panel showing RNA fluorescence in situ hybridization (RNA-FISH) of day 6 *P. cynomolgi* Sz and Hz with probes against *gapdh* (PcyM_1250000) and *hsp70* (PcyM_0515400). Scale bars, 20 μm. Lower panel shows gene expression values (FPKM) for *gapdh* and *hsp70* of individual Hz and Sz samples as determined by RNA-sequencing.

DOI: https://doi.org/10.7554/eLife.29605.002

The following source data and figure supplements are available for figure 1:

**Figure supplement 1.** Comparison with published data.
DOI: https://doi.org/10.7554/eLife.29605.003
**Figure supplement 2.** IFA staining of acetylated H4K8 in *P.cynomolgi* liver stages.
DOI: https://doi.org/10.7554/eLife.29605.004
**Figure supplement 3.** Normalization of gene expression values.
DOI: https://doi.org/10.7554/eLife.29605.005
**Figure supplement 3—source data 1.** Normalization of selected Plasmodium genes.
DOI: https://doi.org/10.7554/eLife.29605.006

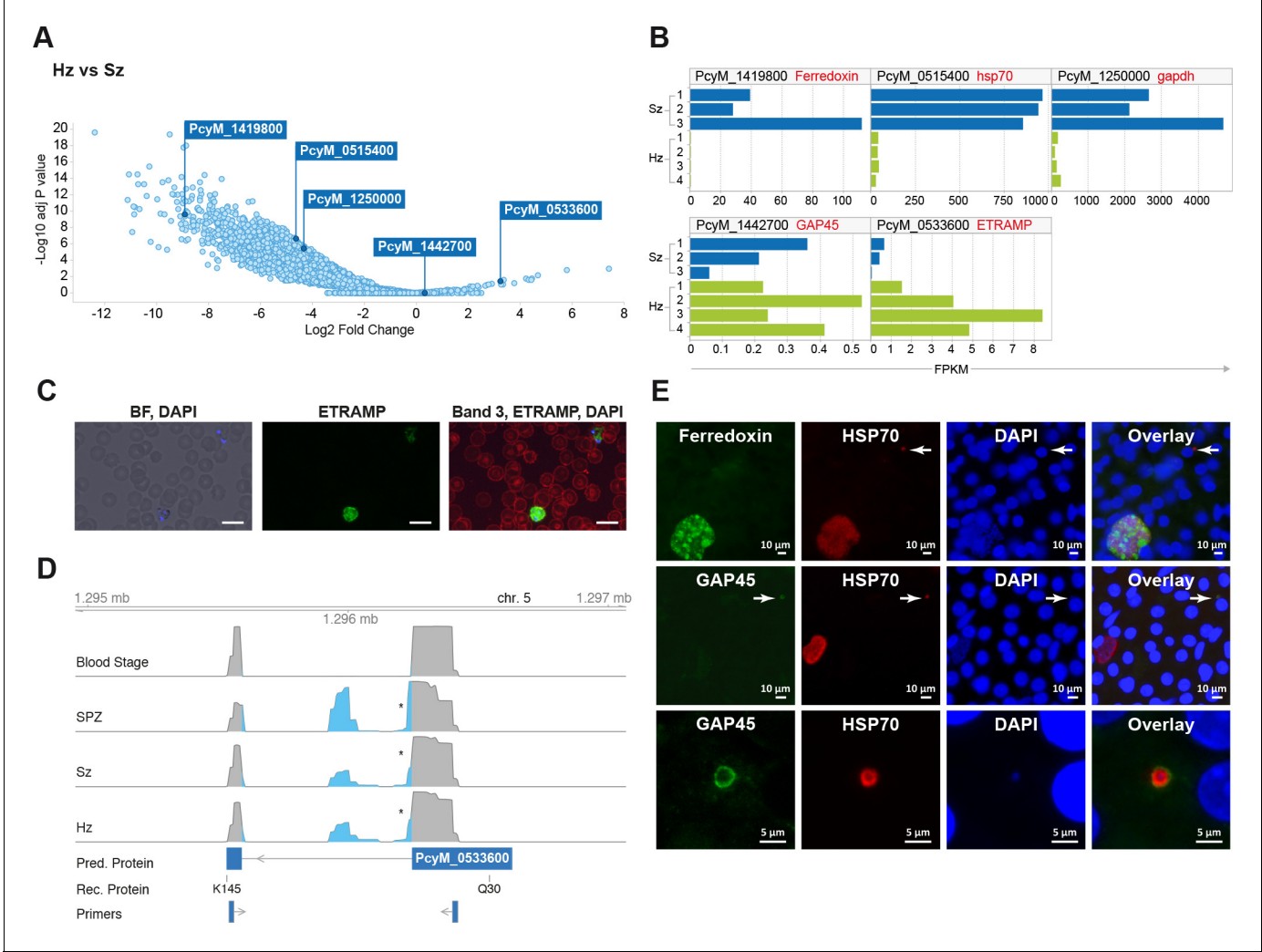

**Figure 2.** Relapsing malaria liver stages display low transcription levels and differ from developing stages. (**A**) Volcano plot showing genes differentially expressed in hypnozoites (Hz, n = 4 biological replicates) versus schizonts (Sz, n = 3 biological replicates). The *y*-axis represents the significance as –log10 FDR-adjusted *p*-values and the *x*-axis represents the expression changes as log2 fold-change of Hz and Sz. Genes used for validation are marked. (**B**) Gene expression values (FPKM) of individual Hz and Sz samples from genes selected for validation. (**C**) Immunofluorescent staining of ETRAMP protein (green), DAPI (blue) and red blood cell (red) in *P. cynomolgi* blood stage parasites. Scale bars 25 μm. (**D**) Genome browser view of the *etramp* gene (PcyM_0533600) showing intron splicing events detected by sequencing of RT-PCR products in Blood stages, Sporozoites (SPZ), Schizonts (Sz) and Hypnozoites (Hz). Retained intron events are highlighted in blue; asterisk shows premature termination codons (PMTs). The predicted protein (Pred. Protein), the recombinant portion of the protein (Rec. Protein) used for antibody production (amino acids Q30-K145), and the positions of the primers used to generate the RT-PCR products are shown. (**E**) Immunofluorescent staining patterns of Ferredoxin (PcyM_1419800), GAP45 (PcyM_1442700), and HSP70 (PcyM_0515400) in day 6 *P. cynomolgi* liver schizonts and hypnozoites. Arrows, hypnozoites. Lower panel shows magnified image of GAP45 stained hypnozoite.

DOI: https://doi.org/10.7554/eLife.29605.007

The following figure supplement is available for figure 2:

**Figure supplement 1.** GAP45 protein expression in day 19 hypnozoite.

DOI: https://doi.org/10.7554/eLife.29605.008

reprocessed the raw sequencing files using the *P. cynomolgi* genome from Pasini *et al.* (*Pasini et al., 2017*) and the data analysis methods that we used in this manuscript. The schizonts data from Cubi *et al.* showed a high correlation of 0.95 and a large consensus between the two replicates, which compared with a slightly lower but equally high correlation (average correlation 0.88) and high overlap between the triplicates profiled in this study (*Figure 1—figure supplement 1*). In stark contrast, while gene expression data reported here showed a high concordance between the four biological

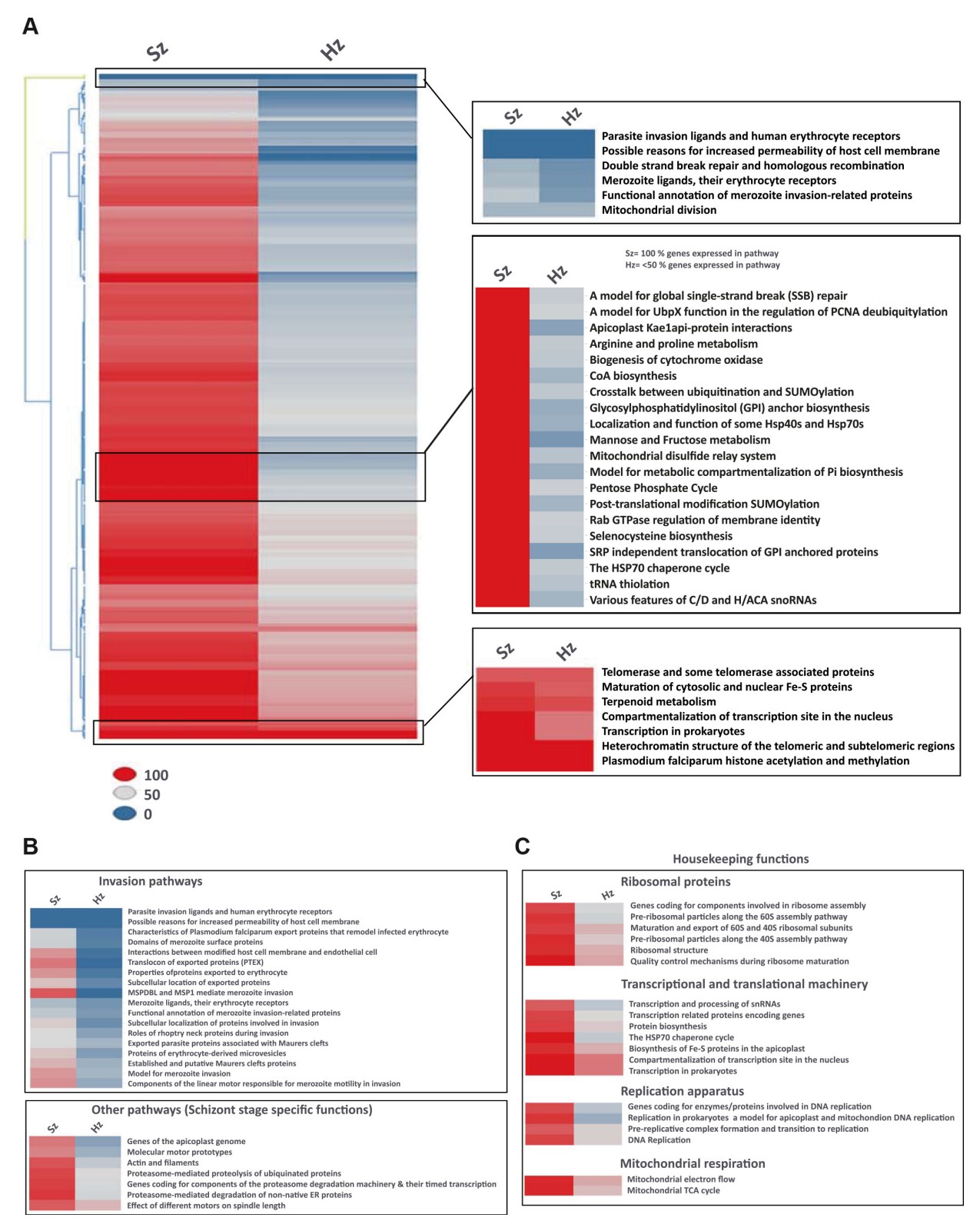

**Figure 3.** Pathway analysis of the malaria liver stages reveals the core biological functions required for hypnozoites maintenance. (**A**) Heat map showing expression of *Plasmodium* pathways in schizonts and hypnozoites. A total of 257 pathways annotated in *P. falciparum* were assigned to *P. cynomolgi* through orthology (see Materials and methods). Pathways where the fraction of genes detected above the threshold of FPKM of 1 is 100% are shown in *Figure 3 continued on next page*

*Figure 3 continued*

red, between 50% and 100% in grey, between 0% and 50% in blue. (**B**) Same as a) but showing only erythrocytic invasion and schizont specific pathways. (**C**) Same as a) but showing house-keeping pathways.

DOI: https://doi.org/10.7554/eLife.29605.009

The following figure supplements are available for figure 3:

**Figure supplement 1.** Liver stage schizont (Sz) and hypnozoite (Hz) gene expression values (FPKM) for pathways associated with quiescence.

DOI: https://doi.org/10.7554/eLife.29605.010

**Figure supplement 2.** Expression of transcription factors in hypnozoites.

DOI: https://doi.org/10.7554/eLife.29605.011

replicates of hypnozoites (average correlation 0.68) and a large overlap of 2804 out of 4198 genes expressed in at least two samples (*Figure 1—figure supplement 1*), the data from Cubi *et al.* showed a lower correlation for the two hypnozoite samples (correlation of 0.38; *Figure 1—figure supplement 1*) and a scarce consensus between the two replicates (204 out of 1147 genes; *Figure 1—figure supplement 1*). Of these 204 genes, 175 overlap with at least one of our hypnozoite samples (*Figure 1—figure supplement 1*), and can thus be considered true hypnozoite transcripts. Compared to the here reported 2804 hypnozoite transcripts, this indicates that many genes and pathways expressed in hypnozoites were not captured in the previous study (*Cubi et al., 2017*).

Although the transcription level in hypnozoites appears to be generally reduced, we found evidence that there is ongoing active transcription in hypnozoites, up to day 7, as demonstrated by the positive staining with antibodies recognizing the acetylated H4K8 protein, a marker of open chromatin (*Gupta et al., 2013*) (*Figure 1—figure supplement 2*). Thus, when compared to proliferating liver schizonts, 'dormant' hypnozoites express only less than half of the parasite genome and the rate of transcription of individual genes also appears to be very low.

## Comparative transcriptomic analysis allows the identification of differential markers of *P. cynomolgi* liver stages

We further explored the liver stage transcriptomes to identify those genes with significantly different expression levels between hypnozoites and schizonts (>2 fold-change absolute value, 10% false discovery rate (FDR); *Supplementary file 5*) (*Figure 2A and B*). Our results indicate that the expression of only a dozen genes might be enhanced in quiescent hypnozoites as compared to growing liver schizonts, while the expression of thousands of genes is significantly lower in hypnozoites than in schizonts. To determine whether protein expression follows the RNA differential expression observed, we selected a few genes that were upregulated in either stage and raised antibodies against recombinant predicted proteins. Using these antibodies, we then performed immunofluorescence analysis (IFA) on cultured liver stages.

Unexpectedly, antibodies against PcyM_0533600 (ETRAMP, amino acids Q30-K145), one of the most up-regulated genes in the hypnozoite samples, failed to detect the protein in day 6 *P. cynomolgi* liver stage parasites. The same antibodies strongly reacted with *P. cynomolgi* blood parasites (*Figure 2C*) but failed to detect the protein in sporozoites (data not shown). Sequence analysis of RT-PCR products from different parasite stages revealed that only blood stage parasites express the predicted full-length PcyM_0533600 mRNA, while alternatively spliced transcripts (including premature stop codons) were found in sporozoite, schizont, and hypnozoite samples (*Figure 2D*), explaining our inability to detect the predicted protein in these stages.

To further validate our dataset, antibodies were raised against three other proteins, PcyM_0515400 (HSP70), PcyM_1419800 (Ferredoxin) (*Miotto et al., 2015*) and PcyM_1442700 (Glideosome Associated Protein GAP45). These antibodies did react with *P. cynomolgi* day six liver stage parasites showing staining of liver schizonts (Ferredoxin), primarily hypnozoites (GAP45) or both schizonts and hypnozoites (HSP70), mirroring precisely the RNA-seq data for these genes (*Figure 2E*). Antibodies against GAP45, an inner membrane complex (IMC) marker (*Kono et al., 2012*) and a member of the glideosome motor complex (*Harding and Meissner, 2014*), stained the periphery of 6 days old hypnozoites (*Figure 2E* middle and lower panels). In contrast, the staining pattern in schizonts was weaker and sparsely distributed (early schizonts) or absent (large mature schizonts) (*Figure 2E* middle panel). These data concur with previous reports describing the

**A**

| Gene ID | Target | Description | Compound | Sz (avg FPKM) | Hz (avg FPKM) |
|---------|--------|-------------|----------|---------------|---------------|
| PcyM_1471000 | FKBP35 | FK506-binding protein (FKBP)-type peptidyl-prolyl isomerase | D44 | 140 | 19 |
| PcyM_1337500 | KRS | Lysine--tRNA ligase (apicoplast) | Cladosporin | 98 | 17 |
| PcyM_1245400 | DXR | 1-deoxy-D-xylulose 5-phosphate reductoisomerase | Fosmidomycin | 211 | 12 |
| PcyM_1264200 | eEF2 | elongation factor 2 | DDD107498 | 124 | 5 |
| PcyM_0526900 | DHFR-TS | bifunctional dihydrofolate reductase-thymidylate synthase | Pyrimethamine | 118 | 4 |
| PcyM_1148700 | DHODH | dihydroorotate dehydrogenase, mitochondrial precursor | DSM265 | 113 | 2 |
| PcyM_1430900 | DHPS | hydroxymethylpterin pyrophosphokinase-dihydropteroate synthetase | Sulfadoxine | 28 | n.d. |
| PcyM_1023600 | PI4K | phosphatidylinositol 4-kinase | KDU691 | 25 | n.d. |
| PcyM_0207400 | PheRS | phenylalanine--tRNA ligase alpha subunit | BRD3444 | 20 | n.d. |
| PcyM_1312800 | ATP4 | P-type ATPase4 | KAE609 | 2 | n.d. |

**B**

| Gene ID | Gene name | Description | Sz (avg FPKM) | Hz (avg FPKM) | Heavy metal chelator |
|---------|-----------|-------------|---------------|---------------|----------------------|
| PcyM_1331900 | CTR2 | copper transporter, putative | 306 | 13 | Neocuproine |
| PcyM_1277100 | CTR1 | copper transporter, putative | 165 | 6 | Neocuproine |
| PcyM_0923300 | MIT1 | CorA-like Mg2+ transporter protein, putative | 207 | 4 | |
| PcyM_1142600 | ZIP1 | zinc transporter protein, putative | 9 | 3 | |
| PcyM_0609400 | ZIPCO | ZIP domain-containing protein putative (ZIPCO) | 402 | 2 | Desferrioxamine (DFO) |
| PcyM_1444100 | VIT | iron transporter, putative | 150 | 1 | Desferrioxamine (DFO) |

**C**

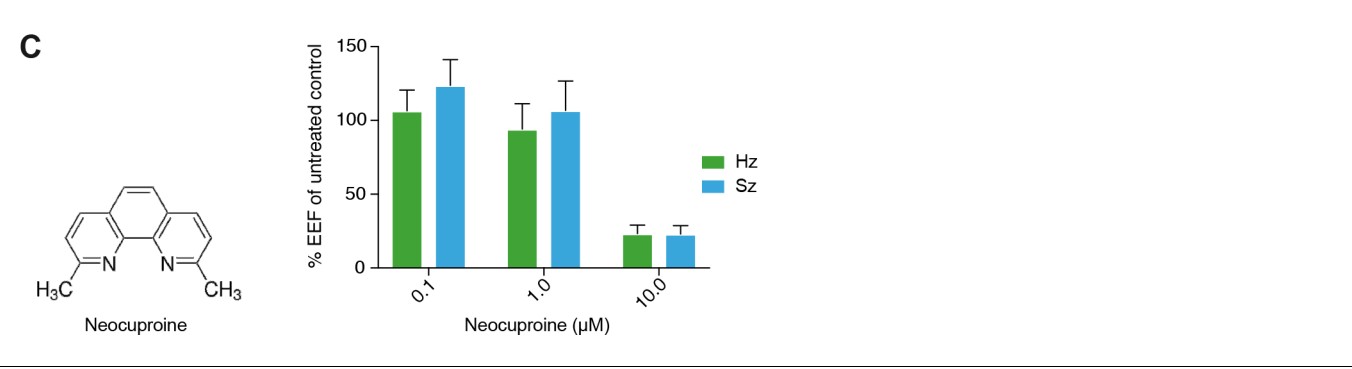

**Figure 4.** Expression of potential malaria drug targets in hypnozoites. (**A**) Table showing the list of known malaria drug targets along their expression levels in the liver stages and the targeting compound. (**B**) Table showing list of putative heavy metal transporters with chelating agents and their expression levels in the *P. cynomolgi* liver stages. (**C**) Structure formula of the copper chelator neocuproine. Dose-dependent effect of day 0–6 neocuproine treatment on *P. cynomolgi* liver stage schizonts (Sz) and hypnozoites (Hz). Bar charts show averaged results of 3 independent assays (7 wells per compound dilution in total) with standard error of the mean (sem). List of Supplemental Tables.
DOI: https://doi.org/10.7554/eLife.29605.012
The following source data is available for figure 4:

**Source data 1.** Neocuproine treatment of liver schizonts and hypnozoites.
DOI: https://doi.org/10.7554/eLife.29605.013

progressive loss of the IMC during conversion of the motile sporozoite into a replication-competent metabolically active liver stage form (*Jayabalasingham et al., 2010*). Interestingly, we could still detect GAP45 in hypnozoites at day 19 (*Figure 2—figure supplement 1*). The long-term presence of GAP45 may be due to low protein turnover in hypnozoites or a functional requirement of this protein for hypnozoite maintenance. Taken together the RNA-FISH and immunofluorescence experiments confirmed the general trends we observed in the RNA-seq dataset and we anticipate that further mining of this gene list will yield differential markers of schizont development and hypnozoite maintenance.

## Hypnozoites express few core pathways including the physiological hallmarks of dormancy

To investigate the physiology of *P. cynomolgi* liver stages, we performed a pathway analysis in schizonts and hypnozoites. Through orthology mapping, *P. cynomolgi* genes were assigned to 257 *Plasmodium falciparum* pathways (*Aurrecoechea et al., 2009*) (*Supplementary file 6*, *7*). Gene ontology and pathway enrichment analyses highlighted that hypnozoites express genes related to translation, RNA processing and epigenetic processes (*e.g.* histone acetylation and methylation) (*Supplementary file 8*). These pathways and processes were also enriched in the schizonts, which however expressed more processes related to the cell nucleus, hinting at the differences in transcriptional activity (*Supplementary file 9*). Schizonts clearly express a much higher number of pathways than hypnozoites (*Figure 3A*). Of all the pathways included in this analysis, only ~34% (88 out of 257 pathways) express more than half of their constituent genes above the threshold of 1 FPKM in the hypnozoite while the equivalent is true for ~91% (233 out of 257) of the pathways in schizonts. In the schizonts, energy and glucose metabolism pathways, such as pentose phosphate cycle enzymes, CoA biosynthesis pathways and mannose/fructose metabolism are all highly expressed with nearly all genes in those pathways detected above 1 FPKM (*Figure 3A*). In contrast, those pathways are nearly absent in the hypnozoite, which is consistent with the quiescence and low metabolism that may be expected in dormant forms.

Interestingly, some but not all erythrocytic invasion pathways are expressed only in schizonts, suggesting that already at day six the parasites express some of the genes required for merozoite function and red blood cell invasion (*Figure 3B*). Hypnozoites mostly express core housekeeping pathways such as those involved with nucleus and chromatin maintenance, transcription, translation and mitochondrial respiration, but no DNA replication enzymes (*Figure 3C*, *Figure 3—figure supplement 1*). Notably, genes known to be required for ATP homeostasis in non-replicating dormant *Mycobacterium tuberculosis* (*Rao et al., 2008*), such as various components of the F0-F1 ATPase complex, are similarly significantly expressed in hypnozoites (*Figure 3—figure supplement 1*). Collectively, our analyses reveal that hypnozoites express pathways previously associated with quiescence and required for the maintenance of chromosome integrity and ATP homeostasis.

## Expression pattern of potential drug targets in *P. cynomolgi* liver stages

In the liver schizonts and hypnozoites transcriptomic dataset, we looked at the expression FPKM values for clinically and chemically validated drug targets (reported in *Figure 4A*). While all drug targets are expressed in the schizonts above the threshold level of 1 FPKM, only a few of them are detectable above this level in the hypnozoites. For example, we could not detect PI4K transcripts in day six hypnozoites while this gene is abundantly expressed in schizonts (*Figure 4A*), which is consistent with previously published data on *Plasmodium* PI4K inhibitors having prophylactic but not radical curative activity in the *P. cynomolgi* model (*Zeeman et al., 2016*). In contrast, the antifolate drug target DHFR is detectable above 1 FPKM in hypnozoites, yet antifolates do not exhibit radical cure in the *P. cynomolgi* model (*Schmidt et al., 1982*). Likewise, DHODH, the target of the clinical candidate DSM265, is detectable in hypnozoites while this compound shows poor activity against hypnozoites in vitro (*Phillips et al., 2015*). Although the *P. cynomolgi* ATP4 ortholog, the clinically validated target of KAE609, is detectable in schizonts at low level, it is not critical as PfATP4 inhibitors are not active in liver stages (*Jiménez-díazDíaz et al., 2014*; *Rottmann et al., 2010*; *Vaidya et al., 2014*). Thus, it appears that function could not be directly inferred from the liver stages expression data.

*Plasmodium* parasite survival and replication depends on the import of nutrients and solutes from its host cell and some transporters have been proposed to be tractable drug targets for malaria (*Hapuarachchi et al., 2017*; *Pain et al., 2016*; *Slavic et al., 2011*; *Weiner and Kooij, 2016*). Consistently, we observe high FPKM values for a broad range of transporters in both liver schizonts (35 putative transporters with FPKM values > 10) and hypnozoites (seven transporters with FPKM values > 10 and 25 transporters with FPKM values > 1 (*Supplementary file 4*, *Supplementary file 10*). Heavy metal homeostasis has been shown to be critical to liver stage (*Kenthirapalan et al., 2016*; *Sahu et al., 2014*; *Stahel et al., 1988*) development and consistently we found several heavy metal transporters to be expressed in all liver stages (*Supplementary file 10*, *Figure 4B*). Remarkably, two

putative copper transporters (PcyM_1331900 and PcyM_1277100) showed high FPKM values for both liver stage schizonts and hypnozoites (highlighted in *Supplementary file 10*, *Figure 4B*), suggesting a role for copper homeostasis in liver stage development and quiescence. To determine whether copper was critical to *P. cynomolgi* liver stages, we treated infected hepatocytes with a copper chelator, neocuproine (*Choveaux et al., 2012*; *Kenthirapalan et al., 2014*). Neocuproine treatment, initiated 1–2 hr after infection with sporozoites and continued for 6 days, indeed showed pronounced cidal effects on the viability of both liver schizonts and hypnozoites (*Figure 4C*) at the highest concentration tested. In one of the three assays we noted a limited effect on hepatocyte viability at this concentration, as concluded from from hepatocyte nuclei counts in the analysis (*Figure 4—source data 1*). These data provide some preliminary chemical validation of the hypothesis that copper homeostasis may be critical for schizonts replication and hypnozoites survival.

## Discussion

Because malaria liver stage parasites are more difficult to culture in vitro, the parasite hepatic life cycle has been neglected and our collective knowledge of those stages remains sparse. The need for new pre-erythrocytic vaccination strategies (*Longley et al., 2015*) and novel drug therapies to combat relapsing malaria parasites (*Campo et al., 2015*), recently fueled much interest for further investigations of the biology of liver stage parasites. As a significant contribution to these efforts, we report here a comprehensive comparative transcriptomics dataset of both developing and dormant liver stage *P. cynomolgi* malaria parasites. Using this dataset, we identified two protein markers that differentiate quiescent from actively dividing parasites and demonstrate that copper homeostasis is critically required for *P. cynomolgi* parasites replication and survival in hepatocytes. It is our hope that through multi-disciplinary collaborative efforts the research community will further mine this dataset to gain further insights in the biology of *Plasmodium* dormancy.

Recently, a first *P. cynomolgi* hypnozoite transcriptomic dataset has been published (*Cubi et al., 2017*), which reports about 120 differentially expressed genes of which 69 are more than 3-fold upregulated, while we report here a much smaller number of upregulated genes in hypnozoites. It is important to note that Cubi *et al.* applied a uniform normalization that assumes that signals from different samples should be scaled to have the same median or average value thus not taking into account the size differences of replicating and dormant liver stages (*Mikolajczak et al., 2015*). This could have potentially biased their comparative analysis towards an over-estimation of the gene expression levels in hypnozoites. In contrast, we applied a group-wise normalization to the expression data in order to keep the expected difference in absolute level of gene expression between schizont and hypnozoite (*Figure 1—figure supplement 3*; *Figure 1—figure supplement 3—source data 1*). Cubi *et al.* proposed that the gene PCYB_102390 (PcyM_1014300 in our dataset) encodes an ApiAP2 transcription factor AP2-Q (for quiescence) which could act as a master regulator of the hypnozoite fate (*Cubi et al., 2017*). However even after normalization, we failed to detect expression of this gene in four hypnozoite samples (*Supplementary file 4*, *Figure 3—figure supplement 2*). Notwithstanding we detected transcripts for nine other Api-AP2 genes in hypnozoites (*Supplementary file 4*, *Figure 3—figure supplement 2*). None of the AP-AP2 genes, including PcyM_1014300, are, however, exclusive for relapsing malarias, as suggested previously (*Cubi et al., 2017*). Only further functional characterization, like the studies that revealed the role of the AP2-G and AP2-G2 genes in gametocyte commitment (*Sinha et al., 2014*), will reveal the possible role of these AP2 transcription factors in hypnozoite identity and survival.

We have previously shown that hypnozoite physiology evolves over time and while PI4 kinase (PI4K) inhibitors are protective when administered shortly after the initial malaria liver infection, they fail to radically cure monkeys when administered several days after parasite inoculation (*Zeeman et al., 2016*). In agreement with our previous reports, we found that at least as early as day six post-infection, the *P. cynomolgi* PI4K gene is not expressed in hypnozoites. The current in vitro liver stage drug assays cannot distinguish compounds only active against developing hypnozoites from those with activity against established hypnozoites (*Zeeman et al., 2016*). The identification of markers specific to established hypnozoites would inform the design of parasites with transgenic reporter genes that would greatly assist the development of in vitro drug screening platforms (*Campo et al., 2015*). We found few upregulated genes in hypnozoites (*Supplementary file 5*) and unfortunately the most highly differentially expressed gene, PcyM_0533600, a member of the

*etramp* family, was not translated in hypnozoites and sporozoites. The presence of such unproductive alternatively spliced transcripts in *Plasmodium* is not uncommon (*Sorber et al., 2011*) and translational repression, including that of a member of the etramp family, UIS4 (*Silvie et al., 2014*), has been shown to be involved in transitions between developmental stages of the life cycle (*Lasonder et al., 2016*). We showed nonetheless that the comparative transcriptomic dataset from this work can help select suitable proteins to produce monoclonal antibodies that differentially label specific liver stages. Further experiments are ongoing to expand the malaria liver stage research toolbox with selective and specific antibodies for replicating and quiescent liver stages.

Dormancy is a physiological response which is relevant to various chronic human infectious diseases and shared by a wide range of pathogens expressing physiological hallmarks characteristic of microbial quiescence. Our dataset suggests that several of these hallmarks are present in the *Plasmodium* hypnozoite—namely the maintenance of membrane potential, ATP biosynthesis and preservation of genome integrity (*Rittershaus et al., 2013*). Indeed, pathway analysis reveals that most mitochondrial electron flow genes and ATP production enzymes are robustly expressed in hypnozoites (*Figure 3—figure supplement 1*). Similarly, nucleus and chromatin maintenance genes are highly expressed in hypnozoites, and while canonical non-homologous end joining (NHEJ) DNA-repair pathways are not present in *Plasmodium* (*Gardner et al., 2002*), we detected most of the homologous recombination repair (HR) enzymes as well as genes required for the maintenance of epigenetics marks (*Figure 3—figure supplement 1*). In addition, the transcriptomics data together with the FISH validation experiments, suggest that hypnozoites display a significant reduction in transcriptional rate both qualitatively and quantitatively which is another hallmark of quiescence (*Figure 1C and D*, and *Supplementary file 4*). All of these physiological hallmarks do theoretically provide proven therapeutic approaches for killing quiescent organisms with the targeting of pathogens' RNA polymerase(s) (*Sala et al., 2010*), proton-motive force enzymes (*Andries et al., 2005*) and DNA repair or epigenetic regulators (*Dembélé et al., 2014*; *Sala et al., 2010*). Establishing selective inhibition of these essential physiological processes in the parasite without significant toxicity to the host cells will be the key challenge for such approaches to be successful.

In order to survive, malaria parasites utilize membrane transport proteins that allow the uptake of nutrients, disposal of waste products and maintenance of ion homeostasis (*Weiner and Kooij, 2016*). While some of these transporters have been implicated in drug resistance, recent experimental work has also supported their potential as anti-malarial drug targets (*Weiner and Kooij, 2016*). Recent evidence has emerged for important roles of heavy metal homeostasis in sporozoite transmission and liver-stage development (*Kenthirapalan et al., 2016*; *Sahu et al., 2014*; *Slavic et al., 2016*; *Stahel et al., 1988*). Iron-deprivation inhibits liver stage growth (*Goma et al., 1995*; *Stahel et al., 1988*) and inactivation of a zinc-iron permease (ZIPCO) was shown to be detrimental for liver stage development (*Sahu et al., 2014*). We report here that *P. cynomolgi* liver stage parasites express transporters for heavy-metals, including copper that in these preliminary experiments seems to be crucially needed for liver stages. Targeting such essential import pathways will again require selective inhibition of the parasite transporters for such approaches to be viable therapeutically.

Taken together the RNA-seq data indicate that drug target liver stage expression is necessary but clearly not sufficient for an inhibitor to show anti-parasitic liver stage activity. Nonetheless, it is worth noting that the 1-deoxy-D-xylulose 5-phosphate reductoisomerase (DXR), the target of Fosmidomycin (*Umeda et al., 2011*), and the Elongation Factor 2 (eEF2), the target of the recently discovered drug candidate DDD107498 (*Baragaña et al., 2015*), are both expressed in the hypnozoite at day 6. This may warrant further investigations of the potential of these compounds for vivax malaria radical cure. Although we did not identify pathways or drug targets specific to hypnozoites, our data collectively show that the hypnozoite expresses a core set of genes required for its basic cellular function. Identifying those essential functions that could be safely targeted with small molecule inhibitors should reveal the Achilles' heel of the elusive hypnozoite.

## Materials and methods

### Ethics statement

Nonhuman primates were used because no other models (in vitro or in vivo) were suitable for the aims of this project. The local independent ethical committee constituted conform Dutch law (BPRC Dier Experimenten Commissie, DEC) approved the research protocol (agreement number DEC# 708) prior to the start and the experiments were all performed according to Dutch and European laws. The Council of the Association for Assessment and Accreditation of Laboratory Animal Care (AAALAC International) has awarded BPRC full accreditation. Thus, BPRC is fully compliant with the international demands on animal studies and welfare as set forth by the European Council Directive 2010/63/EU, and Convention ETS 123, including the revised Appendix A as well as the 'Standard for humane care and use of Laboratory Animals by Foreign institutions' identification number A5539-01, provided by the Department of Health and Human Services of the United States of America's National Institutes of Health (NIH) and Dutch implementing legislation. The rhesus monkeys (*Macaca mulatta*, either gender, age 4–7 years, Indian or mixed origin) used in this study were captive-bred and socially housed. Animal housing was according to international guidelines for nonhuman primate care and use. Besides their standard feeding regime, and drinking water ad libitum via an automatic watering system, the animals followed an environmental enrichment program in which, next to permanent and rotating non-food enrichment, an item of food-enrichment was offered to the macaques daily. All animals were monitored daily for health and discomfort. All intravenous injections and large blood collections were performed under ketamine sedation, and all efforts were made to minimize suffering. Liver lobes were collected from monkeys that were euthanized in the course of unrelated studies (ethically approved by the BPRC DEC) or euthanized for medical reasons, as assessed by a veterinarian. Therefore, none of the animals from which liver lobes were derived were specifically used for this work, according to the 3Rrule thereby reducing the numbers of animals used. Euthanasia was performed under ketamine sedation (10 mg/kg) and was induced by intracardiac injection of euthasol 20%, containing pentobarbital.

### Transgenic *Plasmodium cynomolgi* sporozoite production

Blood stage infections were initiated in rhesus monkeys by intravenous injection of $1 \times 10^6$ *P. cynomolgi* M strain PcyC-PAC-GFP$_{hsp70}$-mCherry$_{ef1\alpha}$ (*Voorberg-van der Wel et al., 2013*) parasites from a cryopreserved stock. To exclude possible wild type contaminant parasites, monkeys were treated with pyrimethamine (1 mg/kg, orally on a biscuit every other day) for 3–4 times starting one day post infection. Parasitemia was monitored by Giemsa-stained smears prepared from a drop of blood obtained from thigh pricks. Animals were trained to voluntarily present for thigh pricks, and were rewarded afterwards. Around peak parasitemia, on two consecutive days, generally at days 11 and 12 post-infection, 9 ml of heparin blood was taken to feed mosquitoes and monkeys were cured from *Plasmodium* infection by intramuscular treatment with chloroquine (7.5 mg/kg) on three consecutive days. Typically ±600 mosquitoes (two to five days old female *Anopheles stephensi* mosquitoes Sind-Kasur strain Nijmegen; Nijmegen UMC St. Radboud, Department of Medical Microbiology) were fed per blood sample using a glass feeder system. Mosquitoes were kept under standard conditions (*Voorberg-van der Wel et al., 2013*). Approximately one week after feeding, oocysts were counted and mosquitoes were given an uninfected blood meal to promote sporozoite invasion of the salivary glands. Mosquitoes that had received blood from the first bleeding ('feed 1') were kept separately and treated independently from mosquitoes that had received blood from the second bleeding ('feed 2').

### Primary hepatocytes

Primary hepatocytes from *Macaca mulatta* or *Macaca fascicularis* were isolated freshly as described before or thawed from frozen stocks and resuspended in William's B medium (*Zeeman et al., 2014*): William's E with glutamax containing 10% human serum (A+), 1% MEM non-essential amino acids, 2% penicillin/streptomycin, 1% insulin/transferrin/selenium, 1% sodium pyruvate, 50 µM β-mercaptoethanol, and 0.05 µM hydrocortisone. Hepatocytes were seeded into collagen coated (5 µg/cm² rat tail collagen I, Invitrogen, Grand Island, NY, USA) 6-well Costar plates at a concentration of

approximately $2.25 \times 10^6$ cells/well. Following attachment, the medium was replaced by William's B containing 2% dimethylsulfoxide (DMSO) to prevent hepatocyte dedifferentiation.

## Sporozoite isolation and inoculation

Two weeks post mosquito feeding on transgenic *P. cynomolgi* M strain infected blood, salivary gland sporozoites were isolated and used for hepatocyte inoculation. Prior to inoculation, hepatocytes were washed in William's B medium followed by sporozoite inoculation at $\pm 2 \times 10^6$ sporozoites per well. Plates were spun at RT at 500 g for 10–20 min and placed in a humidified 37°C incubator at 5% $CO_2$ for 2–3 hr to allow for sporozoite invasion. Medium (William's B) was replaced and incubation continued. Subsequently, infected hepatocytes were cultured with regular (every other day) medium changes until cell sorting. Sporozoite isolations and hepatocyte inoculations from 'feed 1' mosquitoes were performed separately from 'feed 2' mosquitoes.

## Flow cytometry and cell sorting

At day six post sporozoite inoculation hepatocytes were harvested by Trypsin treatment (0.25% Trypsin-EDTA, Gibco, Grand Island, NY, USA). For logistical reasons, samples PAC22F1 and PAC22F2 were cultured for an additional day and were trypsinized at day seven post-inoculation. Cells were washed once with PBS, followed by 3 min incubation in trypsin at 37°C. Complete William's B medium was added to stop the trypsin digestion, cells were collected and washed two times in William's B medium that was diluted 1:5 in William's E to decrease the amount of serum in the samples. Prior to sorting, cells were passed through a 100 µM cell strainer to exclude clumps. First, a sample of uninfected hepatocytes was analysed to enable gate settings. Subsequently, infected hepatocytes were sorted with a BD FACSAria flowcytometer equipped with a 488 nm Coherent Sapphire solid state 20 mW Laser. Data analyses were performed using FlowJo Version 9.4.10 (TreeStar, Inc., Ashland OR, USA). The device was equipped with a 100 µM nozzle for sorting. Gate settings were essentially the same as reported previously, except that an extra gate ('GFPdim') was included to ensure a strict separation of 'GFPlow' and 'GFPhigh' parasites (*Figure 1B*). Sorted samples were collected in 300 µl Trizol (Invitrogen). For the series of experiments relating to this paper, we performed six blood stage infections. In two out of six blood stage infections, parasitemia was low (<0.2%) at the time of mosquito feeding. This resulted in poor sporozoite yields and not enough liver stage forms for FACSsort. The four other infections all resulted in successful liver stage infections with sufficient parasites for FACSsort, with one of the infections used for validation experiments. The events (yield of Sz and Hz) after cell sorting are presented in the existing *Supplementary file 1* and the results of the alignment statistics are shown in the existing *Supplementary file 3*. Collected 'GFPlow' samples contained 1,193–2,713 Hz (on average 1826 Hz); collected 'GFPhigh' samples contained 921–1,245 cells (on average 1,056 Sz). After sorting, tubes were vortexed ±30 s and transferred to a −80°C freezer for storage until RNA extraction. During sorting, small amounts of GFPlow, GFPdim and GFPhigh samples were collected in William's B to analyze the quality of the sort: samples were transferred to a 96 well plate and analyzed using a high-throughput high-content imaging system (Operetta, Perkin-Elmer, Waltham, MA, USA).

## Neocuproine treatment

Following salivary gland dissection of infected *A. stephensi* mosquitoes 50,000 sporozoites were added per well to primary macaque hepatocyte cultures in 96-well plates as described earlier (*Zeeman et al., 2014*). Neocuproine (Sigma , St. Louis, MD, USA, cat. 121908), dissolved in DMSO and subsequently diluted in William's B medium to 10, 1, and 0.1 µM was added in duplicate or triplicate wells to the cultures after sporozoite invasion and incubated with regular refreshments until fixation at day 6. Medium containing DMSO was used as control. Following methanol fixation immunofluorescence analysis was performed and parasites were counted using a high-content imaging system (Operetta; Perkin-Elmer) as reported previously (*Zeeman et al., 2014*).

## Protein and antibody production

An *E. coli* codon optimized gene for PcyM_0533600 (Genscript, China) was synthesized and protein (Q30-K145) was expressed in BL21 cells. The protein was purified using a Ni-IMAC column followed

by gel-filtration/buffer exchange and used to immunize rats (Eurogentec, Belgium). In addition, monoclonal antibodies were raised against selected proteins at Genscript, China.

## Immunofluorescence analysis (IFA)

For IFA validation assays of hepatic stages, collagen coated Cell Carrier-96 well plates (Perkin Elmer) or Permanox Lab-Tek chamber slides (Nunc, Rochester, NY, USA) were seeded with fresh primary rhesus hepatocytes and infected with wild type *P. cynomolgi* M sporozoites following procedures as described above and previously (*Zeeman et al., 2014*). For long-term culture, to enable IFA analysis of day 19 liver stage parasites, matrigel was placed on top of the hepatocytes as previously described (*Dembélé et al., 2014*). At day 6/7 (or day 19) post sporozoite inoculation, cells were briefly fixed in cold methanol followed by three washes in PBS. Infected hepatocytes were blocked in 100 mM glycine for 5 min. at room temperature. After three washes with PBS, cells were incubated for 1–2 hr at room temperature with hybridoma supernatant (undiluted), polyclonal antiserum (1:100) or purified IgG (25 µg/ml) diluted in PBS. Primary antibodies were mouse mAb anti-H4K8ac (Active motif, Carlsbad, CA, USA, #61525, 1:500 dilution), polyclonal rat-anti-ETRAMP (PcyM_0533600, Eurogentec), mouse mAb 1G4E7 against GAP45 (PcyM_1442700, Genscript) and mouse mAb 5B10C7 against Ferredoxin (PcyM_1419800, Genscript). Anti-*P. cynomolgi* HSP70.1 polyclonal rabbit serum (*Zeeman et al., 2014*) was included to detect parasites. Cells were washed three times in PBS and incubated 1–2 hr at room temperature with secondary antibodies diluted in PBS with DAPI. Fluorescein isothiocyanate (FITC)-labeled goat anti-rabbit IgG (Kirkegaard and Perry Laboratories, 1:200), FITC-labeled goat anti-mouse IgG (Kirkegaard and Perry Laboratories, 1:200), Alexa-594 labeled chicken anti-mouse IgG (Invitrogen, 1:2000), or Alexa-594 labeled chicken anti-rabbit IgG (Invitrogen, 1:2000) were used as secondary antibodies. Following three washes with PBS, mounting was performed with CITIFLUOR AF1 (Agar Scientific, Belgium). Images were taken using a Nikon Microphot FXA fluorescence microscope equipped with a DS-5M digital camera or with a Leica DMI6000B inverted fluorescence microscope equipped with a DFC365FX camera. For IFA staining of blood stage parasites, blood smear preparations of *P. cynomolgi* infected red blood cells were fixed with methanol. Primary and secondary antibodies were diluted in 1% FCS/PBS and each staining was for 1 hr at room temperature. Primary antibodies were polyclonal rat-anti-ETRAMP (PcyM_0533600) at 25 µg/ml and rabbit anti-Band three monoclonal antibody (Abcam ab108414, 1:100). Secondary antibodies were Alexa-594 labeled chicken anti-rabbit IgG (Invitrogen, Grand Island, NY, USA, 1:2000) and mouse serum adsorbed FITC-labeled goat anti-rat IgG (Kirkegaard and Perry Laboratories, Gaithersburg, MD, USA, 1:200); DAPI was included. Slides were rinsed in PBS (4-5x) and mounted with CITIFLUOR AF1. Images were taken using a Leica DMI6000B inverted fluorescence microscope equipped with a DFC365FX camera.

## RNAscope in situ hybridization

*P. cynomolgi* M infected primary rhesus hepatocytes cultured for 6 days in CellCarrier-96 well plates (Perkin-Elmer, Waltham, MA, USA) were fixed for 30 min. at RT in 4% paraformaldehyde in PBS (Affymetrix, Cleveland, Ohio, USA), dehydrated and stored at −20°C until further processing. RNA in situ detection was performed using the RNAscope Multiplex Kit (Advanced Cell Diagnostics, Newark, CA, USA) according to the manufacturer's instructions. RNAscope probes used were: *gapdh* (PcyM_1250000, region 113–997) and *hsp70* (PcyM_0515400, region 606–1837). Following the RNA-FISH protocol, IFA was performed using rabbit anti-PcyHSP70 to stain the parasites as described above. Z-Stack images were acquired on the Operetta system (Perkin-Elmer) using a 40x objective NA 0.95 and maximum projections are shown.

## RNA sequencing

Total RNA was isolated from five different samples of FACS-sorted small parasite infected cells (GFP-low, *e.g.* hypnozoites) and 5 samples of FACS-sorted large parasite infected cells (GFP-high, *e. g.* liver schizonts). All samples were stored in TRIzol (Invitrogen) and total RNA extracted using the Direct-zol RNA MiniPrep Kit (Zymo Research, Irvine, CA, USA) including on-column DNase digestion according to the manufacturer's instructions. RNA amplification was performed using the TargetAmp 2-Round aRNA Amplification Kit 2.0 (Epicentre, Madison, Wisconsin, USA). The quality of the RNA samples (before and after the amplification) was assessed with the RNA 6000 Pico and Nano

kits using the Bioanalyzer 2100 instrument (Agilent Technologies, Santa Clara, CA, USA). RNA-seq cDNA libraries were prepared from the amplified RNA samples using the TruSeq mRNA Sample Prep kit v2 (Illumina, San Diego, CA, USA). The quality of the cDNA libraries was assessed with the Bioanalyzer 1000 DNA kit (Agilent Technologies). RNA-seq cDNA libraries were then sequenced in paired-end mode, 2 × 76 bp, using the Illumina HiSeq2500 platform. Read quality was assessed by running FastQC (version 0.10) on the FASTQ files. Sequencing reads showed high quality, with a mean Phred score higher than 30 for all base positions. Over 857 million 76-base-pair (bp) paired-end reads were used for the bioinformatics analysis. Reads from each sample were aligned to a genomic reference composed of the combination of the malaria parasite *Plasmodium cynomolgi* M strain genome and one of the following host genomes: *Macaca mulatta* (*Zimin et al., 2014*) (http://www.unmc.edu/rhesusgenechip/), and *Macaca fascicularis* (http://www.ncbi.nlm.nih.gov/assembly/GCF_000364345.1/) (*Supplementary file 2*). *Supplementary file 3* provides specification on which host genome was considered for each sample and the read alignment statistics. Reads mapping to the parasite genome was used to quantify gene expression by using the Exon Quantification Pipeline (EQP) (*Schuierer and Roma, 2016*). On average, a range of 38% (minimum) to 84% (maximum) of total reads were mapped to the parasite and host genomes, and between 17% and 65% were aligned to the parasite and host exons (expressed reads). A QC inspection of the aligned sequencing reads showed an expected coverage bias towards the 3' end of the transcripts that is due to the use of the amplification kit. Based on the alignment statistics, we decided to exclude two Sz samples and one Hz sample from further analyses (*Supplementary file 3*). Genome and transcript alignments were used to calculate gene counts based on the *P. cynomolgi* M strain gene annotation (Pcynom M_v2, *Pasini et al., 2017*) provided by the BPRC and the Wellcome Trust Sanger Institute.

Gene raw counts represent the total number of reads aligned to each gene. These values were normalized using the following four-stage approach (*Figure 1—figure supplement 3*; *Figure 1—figure supplement 3—source data 1*). First, gene raw counts were divided by the total number of mapped reads for each sample and multiplied by one million to obtain Counts Per Million (CPM) to account for varying library sizes (library size normalization). In a given sample, one CPM indicates that a specific gene was detected by one read out of one million of mapped reads. Second, a further normalization of the CPMs based on the BioConductor package DESeq2 (*Love et al., 2014*) was performed for the samples of each stage separately to account for the variation of #parasite-cells/#host-cells fraction within one stage (group-wise normalization). Third, an adjustment of mean expression ratio between schizont and hypnozoite samples was computed by using host expression values to further account for the difference in cell size and RNA amount per cell which is expected between the schizont and the hypnozoite liver forms (host normalization). The host normalized counts were further divided by the gene length in kb to obtain the Fragments Per Kilobase per Million values (FPKM) (gene length normalization). The host normalized gene expression values (available in *Supplementary file 4*) were also used to identify differences in gene expression between the schizont and the hypnozoite samples using the BioConductor package DESeq2 (*Love et al., 2014*). *Supplementary file 5* shows the list of genes that are differentially expressed between the liver schizonts and the hypnozoites along with the log2 fold changes and *p*-values after Benjamini-Hochberg false discovery rate (FDR) correction for multiple hypothesis testing.

## Orthology and pathway analysis

In order to annotate the *Plasmodium cynomolgi* proteome, we performed an extensive orthology analysis that included the following proteomes in addition to *P. cynomolgi* M strain: *P. falciparum 3D7*, *P. berghei ANKA*, *P. knowlesi H*, *P. vivax Sal1*, *P. yoelii yoelii 17X*, *H. sapiens*, *D. melanogaster*, *M. musculus*, *R. norvegicus*, and *S. cerevisiae*. The *Plasmodia* proteomes were obtained from PlasmoDB (http://PlasmoDB.org/) version 26 (*Aurrecoechea et al., 2009*), the other proteomes from UniProt (release 2015_12) (*Bateman et al., 2015*). Our orthology analysis is based on the OrthoMCL methodology but implemented in-house to work with our local high-performance computing environment. Conceptually, this comprised the following steps: (1) alignment of all protein sequences against each other with blastp (*Altschul et al., 1990*); (2) calculation of the percent match length by determining all amino acids participating in any HSP between two proteins divided by the length of the shorter protein; (3) filtering out of the blast results with a percent match length below 50% or an E-value above $10^{-5}$; (4) determination of potential orthologs and paralogs and their normalized E-values; (5) clustering of the resulting weighted similarity graph with MCL. See Fischer 2011 *et al*

(*Fischer et al., 2011*) for more details, and Figure 6.12.1 contained within for an overview. The obtained groups of proteins were used to propagate protein annotations from other species to *P. cynomolgi*. Using this approach, we were able to group a total of 6,040 *P. cynomolgi* proteins (86% of the total 7030 proteins) with at least one protein from another species (*Supplementary file 6*), and 2295 (33%) *P. cynomolgi* proteins were linked to 257 malaria pathways mapped from PlasmoDB (*Supplementary file 7*). For the identification of pathways that are expressed in the liver stages, we used a stringent cut-off to focus only on those genes whose expression is consistent across replicates (>1 FPKM in at least two replicates). This resulted into 2748 genes and 88 pathways expressed in 2/4 Hz replicates, and 5323 genes and 233 pathways expressed in 2/3 Sz replicates.

## Pathway and Gene Ontology enrichment analyses

Gene sets were collected from two sources: PlasmoDB (*Aurrecoechea et al., 2009*) and Gene Ontology (*Ashburner et al., 2000*; *The Gene Ontology Consortium, 2017*). The gene sets from PlasmoDB mostly correspond to 'Metabolic pathways', whereas the gene sets from the Gene Ontology correspond to general organizational principles of biology (such as 'translation'). Many of the pathways from PlasmoDB are manually curated, whereas large parts of the annotations in the Gene Ontology are derived and propagated from one species to another by algorithms. The gene sets were mapped by orthology to *Plasmodium cynomolgi*. We employed two standard approaches to determine the relevance of gene sets with respect to our RNAseq data: 1) overrepresentation analysis *via* a hypergeometric test; and 2) Kolmogorov-Smirnov test, as proposed in the original GSEA publication (*Subramanian et al., 2005*). The main differences between the two approaches is that the first one requires a predetermined criterion to select genes of interest in which overrepresented annotations are to be determined; the second does not need any such cut-off, as the test statistic is based on a ranking of all genes in the experiment.

For the enrichment analyses, we applied several criteria of increasing stringency to select stage-specific genes of interest from our RNA-seq experiment:

- All genes within a certain stage that are expressed with at least 1 FPKM in at least two samples in that stage (*e.g.* Hz or Sz).
- All genes within a certain stage that are expressed with at least $P_{25}$ FPKM in at least two samples in that stage, where $P_{25}$ is the 25th percentile (first quartile) of the expression of the pooled samples of that stage (*e.g._Hz_q1 or Sz_q1).
- All genes within a certain stage that are expressed with at least $P_{75}$ FPKM in at least two samples in that stage, where $P_{75}$ is the 75th percentile (third quartile) of the expression of the pooled samples of that stage (*e.g.* Hz_q3 or Sz_q3);
- All genes that satisfy criterion two in a stage but in no other stage (*e.g.* Hz_q1_specific or Sz_q1_specific).
- All genes that satisfy criterion three in a stage but in no other stage (*e.g.* Hz_q3_specific or Sz_q3_specific).

Genes satisfying the criteria above were determined for all stages and used as input for an overrepresentation analysis. The results for the hypnozoite samples are displayed in *Supplementary file 8*, while those for the schizonts are in *Supplementary file 9*.

## Targeted amplification and sequencing of the *etramp* gene

Blood stage, sporozoite, schizont and hypnozoite RNA samples were reverse transcribed using the High Capacity RNA-to-cDNA Kit (#4368814, Thermo Scientific, Waltham, MA, USA). The *etramp* gene (PcyM_0533600) was amplified in all the samples using the Phusion DNA Polymerase kit (#F530, Thermo Scientific) with the following primers: ACTCCTTGGTGGTGCCTTAG (FWD); TGCGGGGCCCTTATCTTT (REV). The Ovation Low complexity Sequencing System kit (#9092–256, NuGEN, San Carlos, CA, USA) was used to prepare the sequencing libraries. Libraries were multiplexed and sequenced in paired-end mode, at a read length of 2 × 300 bp, using the MiSeq platform (Illumina). The resulting FASTQ files were demultiplexed and aligned against the *P. cynomolgi* M strain genome (Pcynom M_v2, unpublished) using STAR version 2.5.2a (Dobin A, Davis CA, Schlesinger F, Drenkow J, Zaleski C, Jha S and *Dobin et al., 2013*) for the detection of the amplified regions. The Integrative Genomics Viewer (IGV) (James T. Robinson, Helga Thorvaldsdóttir, Wendy Winckler, Mitchell *Robinson et al., 2011*) version 2.3.75 was used to visualize the aligned reads in

the genome context. The etramp gene view presented in *Figure 2D* was generated using the R/Bio-conductor GViz package (*Hahne and Ivanek, 2016*).

## Comparison with published data

Published expression data of *P. cynomolgi* liver stages (*Cubi et al., 2017*) were downloaded from the EMBL-EBI European Nucleotide Archive [ENA: PRJEB18141; Sample group: ERS1461774] and compared to our RNA-seq data. It was not possible to compare the gene lists from Cubi *et al.* directly with the genes from this manuscript because the two studies used different *P. cynolmolgi* reference genomes and gene annotation files. The downloaded Fastq files of Cubi *et al.* were thus processed with the genome reference and annotation files from (*Pasini et al., 2017*), the same RNA-seq analysis pipeline, and the same normalization method as was used for our data set and which is described in the 'RNA sequencing' paragraph above. The correlation plots shown in *Figure 1—figure supplement 1A and B* were generated on log10 normalized CPMs after the addition of a pseudo-count of 0.1. The Venn diagram plots were generated on genes expressed above the cut-off of 1 FPKM in the hypnozoite and schizont samples, and drawn using the on line tool available at the following website: http://bioinformatics.psb.ugent.be/webtools/Venn/.

## Data availability

The raw RNA-sequencing reads are available in the NCBI Short Read Archive (https://www.ncbi.nlm.nih.gov/sra) under accession number SRP096160.

# Acknowledgements

We thank Giovanni D'Ario for bioinformatic support, Els Klooster, Richard Vervenne, Nicole van der Werff and Niels Beenhakker for technical assistance and Francisca van Hassel for preparing graphical representations, and Martin Beibel for statistical support. We are grateful to the mosquito breeding facilities in Nijmegen for provision of *Anopheles stephensi* mosquitoes and Matt Berriman and Thomas Dan Otto from the Wellcome Trust Sanger Institute for access to the *P. cynomolgi* genome prior to publication.

# Additional information

### Competing interests

Guglielmo Roma, Devendra Kumar Gupta, Sven Schuierer, Florian Nigsch, Walter Carbone, Boon Heng Lee, Judith Knehr, Bernd Kinzel, Pablo Bifani, Ghislain M C Bonamy, Tewis Bouwmeester, Thierry Tidiane Diagana: Employed by and/or shareholder of Novartis Pharma AG. The other authors declare that no competing interests exist.

### Funding

| Funder | Author |
| --- | --- |
| Bill and Melinda Gates Foundation | Guglielmo Roma<br>Clemens H M Kocken<br>Thierry Tidiane Diagana |
| Medicines for Malaria Venture | Clemens H M Kocken<br>Thierry Tidiane Diagana |
| Wellcome | Clemens H M Kocken |
| Wellcome | Thierry Tidiane Diagana |

The funders had no role in study design, data collection and interpretation, or the decision to submit the work for publication.

### Author contributions

Annemarie Voorberg-van der Wel, Conceptualization, Data curation, Formal analysis, Validation, Investigation, Visualization, Methodology, Writing—original draft, Writing—review and editing;

Guglielmo Roma, Conceptualization, Resources, Formal analysis, Supervision, Funding acquisition, Investigation, Methodology, Writing—original draft, Writing—review and editing; Devendra Kumar Gupta, Data curation, Formal analysis, Validation, Investigation, Visualization, Methodology, Writing—original draft, Writing—review and editing; Sven Schuierer, Florian Nigsch, Software, Formal analysis, Investigation, Visualization, Methodology, Writing—original draft, Writing—review and editing; Walter Carbone, Formal analysis, Validation, Visualization, Methodology, Writing—review and editing; Anne-Marie Zeeman, Boon Heng Lee, Sam O Hofman, Bart W Faber, Ghislain M C Bonamy, Formal analysis, Investigation, Methodology, Writing—review and editing; Judith Knehr, Data curation, Formal analysis, Validation, Investigation, Methodology, Writing—review and editing; Erica Pasini, Resources, Data curation, Formal analysis, Investigation, Writing—review and editing; Bernd Kinzel, Resources, Formal analysis, Investigation, Methodology, Writing—review and editing; Pablo Bifani, Formal analysis, Investigation, Writing—review and editing; Tewis Bouwmeester, Resources, Formal analysis, Investigation, Writing—review and editing; Clemens H M Kocken, Thierry Tidiane Diagana, Conceptualization, Resources, Supervision, Funding acquisition, Investigation, Methodology, Writing—original draft, Project administration, Writing—review and editing

### Author ORCIDs
Annemarie Voorberg-van der Wel (iD) http://orcid.org/0000-0001-9403-0515
Guglielmo Roma (iD) http://orcid.org/0000-0002-8020-4219
Walter Carbone (iD) http://orcid.org/0000-0001-6150-8295
Thierry Tidiane Diagana (iD) http://orcid.org/0000-0002-8520-5683

### Ethics
Animal experimentation: Ethics statement Nonhuman primates were used because no other models (in vitro or in vivo) were suitable for the aims of this project. The local independent ethical committee constituted conform Dutch law (BPRC Dier Experimenten Commissie, DEC) approved the research protocol (agreement number DEC# 708) prior to the start and the experiments were all performed according to Dutch and European laws. The Council of the Association for Assessment and Accreditation of Laboratory Animal Care (AAALAC International) has awarded BPRC full accreditation. Thus, BPRC is fully compliant with the international demands on animal studies and welfare as set forth by the European Council Directive 2010/63/EU, and Convention ETS 123, including the revised Appendix A as well as the 'Standard for humane care and use of Laboratory Animals by Foreign institutions' identification number A5539-01, provided by the Department of Health and Human Services of the United States of America's National Institutes of Health (NIH) and Dutch implementing legislation. The rhesus monkeys (Macaca mulatta, either gender, age 4-7 years, Indian or mixed origin) used in this study were captive-bred and socially housed. Animal housing was according to international guidelines for nonhuman primate care and use. Besides their standard feeding regime, and drinking water ad libitum via an automatic watering system, the animals followed an environmental enrichment program in which, next to permanent and rotating non-food enrichment, an item of food-enrichment was offered to the macaques daily. All animals were monitored daily for health and discomfort. All intravenous injections and large blood collections were performed under ketamine sedation, and all efforts were made to minimize suffering. Liver lobes were collected from monkeys that were euthanized in the course of unrelated studies (ethically approved by the BPRC DEC) or euthanized for medical reasons, as assessed by a veterinarian. Therefore, none of the animals from which liver lobes were derived were specifically used for this work, according to the 3Rrule thereby reducing the numbers of animals used. Euthanasia was performed under ketamine sedation (10 mg/kg) and was induced by intracardiac injection of euthasol 20%, containing pentobarbital.

### Decision letter and Author response
Decision letter https://doi.org/10.7554/eLife.29605.030
Author response https://doi.org/10.7554/eLife.29605.031

# Additional files

## Supplementary files

• Supplementary file 1. *P. cynomolgi* samples used for RNAseq, *related to Figure 1*
DOI: https://doi.org/10.7554/eLife.29605.014

• Supplementary file 2. Reference genomes used in the analysis, *related to Figure 1*
DOI: https://doi.org/10.7554/eLife.29605.015

• Supplementary file 3. Alignment statistics per sample, *related to Figure 1*
DOI: https://doi.org/10.7554/eLife.29605.016

• Supplementary file 4. Gene expression values (FPKM after host normalization) generated by RNA-seq of hypnozoite (Hz) and liver schizont (Sz) samples, *related to Figure 1*
DOI: https://doi.org/10.7554/eLife.29605.017

• Supplementary file 5. List of differentially expressed genes between hypnozoite and liver schizont samples passing the cut-off of >2 fold change absolute value and a 10% false discovery rate, *related to Figure 2*
DOI: https://doi.org/10.7554/eLife.29605.018

• Supplementary file 6. Orthology groups obtained from the mapping of *P. cynomolgi* proteins to the proteomes of the following species: *P. falciparum* 3D7, *P. berghei* ANKA, *P. knowlesi* H, *P. vivax* Sal1, *P. yoelii yoelii* 17X, *H. sapiens*, *D. melanogaster*, *M. musculus*, *R. norvegicus*, and *S. cerevisiae*, *related to Figure 3*
DOI: https://doi.org/10.7554/eLife.29605.019

• Supplementary file 7. *P. cynomolgi* proteins linked to malaria pathways mapped from PlasmoDB, *related to Figure 3*
DOI: https://doi.org/10.7554/eLife.29605.020

• Supplementary file 8. Enrichment for genes expressed in the hypnozoite samples (Hz). Gene sets from the Gene Ontology (GO) and PlasmoDB were used and mapped to *P. cynomologi* identifiers. *p* is the adjusted *p*-value (Benjamini-Hochberg), *related to Figure 3*
DOI: https://doi.org/10.7554/eLife.29605.021

• Supplementary file 9. Enrichment for genes expressed in the schizont samples (Sz). Gene sets from the Gene Ontology (GO) and PlasmoDB were used and mapped to *P. cynomologi* identifiers. *p* is the adjusted *p*-value (Benjamini-Hochberg), *related to Figure 3*
DOI: https://doi.org/10.7554/eLife.29605.022

• Supplementary file 10. Gene expression values (>1 FPKM after host normalization) for putative transporters generated by RNA-seq of hypnozoite (Hz) and schizont (Sz) samples, *related to Figure 4*
DOI: https://doi.org/10.7554/eLife.29605.023

• Transparent reporting form
DOI: https://doi.org/10.7554/eLife.29605.024

## Major datasets

The following dataset was generated:

| Author(s) | Year | Dataset title | Dataset URL | Database, license, and accessibility information |
|---|---|---|---|---|
| Annemarie Voorberg-van der Wel, Guglielmo Roma, Devendra Kumar Gupta, Sven Schuierer, Florian Nigsch, Walter Carbone, Anne-Marie Zeeman, Boon Heng Lee, Sam O. Hofman, Bart W. Faber, Judith Knehr, Erica M. Pasini, Bernd Kinzel, Pablo Bifani, Ghislain M. C. Bonamy, Tewis Bouwmeester | 2017 | Malaria Liver Stages Transcriptome | https://www.ncbi.nlm.nih.gov/sra/?term=SRP096160 | Publicly available at the NCBI Short Read Archive (accession no: SRP096160). |

The following previously published dataset was used:

| Author(s) | Year | Dataset title | Dataset URL | Database, license, and accessibility information |
|---|---|---|---|---|
| Cubi R, Vembar SS, Biton A, Franetich JF, Bordessoulles M, Sossau D, Zanghi G, Bosson-Vanga H, Benard M, Moreno A, Dereuddre-Bosquet N, Le Grand R, Scherf A, Mazier D | 2017 | Transcriptomic analysis of Plasmodium relapsing species identifies potential regulators of hypnozoite commitment and maintenance | https://www.ebi.ac.uk/ena/data/view/ERS1461774 | EMBL-EBI European Nucleotide Archive [ENA: PRJEB18141; Sample group: ERS1461774] |

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
