## [Decision Letter]

[Editors’ note: this article was originally rejected after discussions between the reviewers, but the authors were invited to resubmit after an appeal against the decision.]

Thank you for submitting your work entitled "A comparative transcriptomic analysis of replicating and dormant liver stages of a relapsing malaria parasite" for consideration by *eLife*. Your article has been reviewed by three peer reviewers, and the evaluation has been overseen by a Reviewing Editor and a Senior Editor. The reviewers have opted to remain anonymous.

Our decision has been reached after consultation between the reviewers. Based on these discussions and the individual reviews below, we regret to inform you that your work will not be considered further for publication in *eLife*.

Summary:

Voorberg-van der Wel and colleagues describe a comprehensive analysis of transcriptomes of dormant *Plasmodium cynomolgi* liver stages isolated from infected monkey hepatocytes. This study is of interest because relapsing malaria parasites, such as *P. cynomolgi*, can exist as non-replicating dormant form of the parasite in the liver called hypnozoite that can reactivate and initiate relapsing malaria infection. *P. cynomolgi* is an excellent model system for the study of the human malaria caused by *Plasmodium vivax* hypnozoite owing to the degree of biological and genomic similarity between the species. The information contained in the manuscript provides a valuable set of tools and genomic resource of dormant malaria liver parasites that will be instrumental for the identification of drug targets for hypnozoites. The datasets generated are cautiously curated and analyzed with a clear explanation of the methods used. However, enthusiasm for the manuscript is reduced because recently Cubi and colleagues reported on the first transcriptome of hypnozoites and schizonts from infected monkey hepatocytes and optimized RNA‐seq analysis. The current study does not significantly depart from this recent report.

*Reviewer #1:*

Several comprehensive transcriptomic analyses have been performed and reported on all malaria stages, however this has been more problematic for the hardly accessible hypnozoite stage. One study has recently been published that reports the first transcriptome of hypnozoites and schizonts from infected monkey hepatocytes and optimized RNA‐seq analysis. The authors applied a laser dissection microscopy protocol to isolate individual *Plasmodium cynomolgi* hypnozoites Cubi R. et al., Cell Microbiol. 2017 Aug;19(8). Voorberg-van der Wel et al., work does not significantly depart from Cubi R et al., work but yet fills an important gap of knowledge on parasites dormancy. The problem of relapses due to dormant stages of *Plasmodium vivax* is among the most relevant aspects of malaria infection, which compromises all the efforts for eradication. Voorberg-van der Wel et al., provide a comprehensive transcriptomics resource of dormant and replicating malaria liver parasites that will be instrumental to the identification of drug targets for hypnozoites. The datasets generated are cautiously curated and analyzed with a clear explanation of the methods used.

The authors report a considerably reduced use of 34% of *Plasmodium* physiological pathways, (91% are expressed in replicating schizonts). Importantly mitochondrial respiration for ATP homeostasis and maintenance of genome integrity are kept active. RNA fluorescence in situ hybridization allows a robust comparison of gene expression in schizonts and hypnozoites.

Overall the results are more stringently interpreted than in Cubi R. et al. with only a dozen genes enhanced in quiescent hypnozoites while the expression of thousands of genes is repressed compared the liver schizonts. The authors identified two protein markers that differentiate quiescent from actively dividing parasites but actually failed to identify a hypnozoite specific marker. For example GAP45 is expressed in all invasive and motile stages of the malaria parasites include late liver stage schizonts.

This work is not entirely novel and preliminary but should be of interest to readers of *eLife* who are interested in malaria and infectious diseases in general.

1) The use of primaquine or tafenoquine as perturbators of the transcriptome would be a relevant and informative extension of this work.

2) The data on copper chelator neocuproine was cidal to all liver stage parasites is preliminary and would deserve more robust investigation towards eradication of hypnozoites.

*Reviewer #2:*

The manuscript describes the transcriptomes of *P. cynomolgi* liver stages grown in primary Simian hepatocytes. This study is of interest because relapsing malaria parasites, such as *P. cynomolgi*, form a small, non-replicating dormant form of the parasite in the liver called hypnozoite that can reactivate and initiate relapsing malaria infection. The authors report the transcriptomes of replicating schizonts and persisting hypnozoites, which they are able to differentially separate using FACS sorting and a transgenic parasite strain expressing GFP. They identify several pathways expressed in the hypnozoite and validate expression of a few genes both by RNA FISH and translation at the protein level using antibodies. Interestingly, two copper transporters were robustly expressed in the hypnozoite as well as schizont. Treating cultures with a copper chelator reduced the number of hypnozoites and schizonts in a dose-dependent manner when dosed prophylactically. While the work presented is important to the malaria field, the authors do not identify any transcripts specific to the hypnozoite. Furthermore, a *P. cynomolgi* hypnozoite transcriptome has already been published recently (Cubi et al. 2017). Therefore, the work might be suitable for a more specialized journal.

1) To identify differentially expressed genes in *P. cynomolgi* schizonts compared to hypnozoites the authors normalized the RNAseq FPKM's to the amount of host RNA present in the host hepatocyte because schizonts have many more genome copies and as such many more transcriptionally active units than hypnozoites. There are three major concerns with this method. 1) Schizonts will all have differing numbers of transcriptionally active units dependent on maturation stage. 2) It is expected that dramatic changes will occur between day 6 and 7 timepoints of analysis) during the schizont life cycle as the parasite prepares for segmentation and cytokinesis. 3) Most importantly, the quantity of host hepatocyte RNA is expected to be dramatically different in hypnozoite-infected hepatocyte versus a massive schizont-infected hepatocyte, with the latter containing much less host RNA. In consequence, any claims about transcriptional repression in hypnozoites cannot be supported by the data and should be removed from the manuscript or a convincing analytical strategy should be presented that fully supports the conclusions.

2) The authors next conduct FISH to support their differential transcript abundance analysis conclusions drawn from RNAseq data. There are two issues with the FISH conclusions. First, FISH is not a quantitative method and in consequence cannot validate quantitative conclusions from RNAseq. Second, even if FISH would allow quantitative analysis, the schizont versus hypnozoite comparison is problematic in that the former contains thousands of transcriptionally active genome copies whilst the latter contains one active genome copy.

3) Do the genes listed in Supplementary file 5 meet any criteria for being differentially expressed? For example, an FDR or FC cutoff? If not, this analysis should be included. Do the authors expect the reported number of genes to be expressed in the liver stages of the parasite and are the transcripts not identified in either liver form expressed in blood stage schizonts?

4) For the *Plasmodium* pathway expression analysis, were any of the pathways determined to be significantly expressed? That is, are the pathways identified in the hypnozoite significantly expressed in hypnozoites compared to other pathways and are those pathways differentially present compared to schizonts? The statistical significance of the enriched pathways should be determined and reported.

5) Much of the authors’ results are contradictory to a recently published *P. cynomolgi* liver stage transcriptome study (Cubi et al., 2017). It is important to give an honest assessment regarding the root causes for these discrepancies. Also move the comparative analysis into the Results section and discuss in the Discussion section.

*Reviewer #3:*

This is an excellent paper that describes a significant advance in malaria relapse biology, through the description of the transcriptome of replicating and dormant liver stages of the monkey malaria parasite *Plasmodium cynomolgi*. Although this is not the first study to undertake a transcriptome analysis of purported hypnozoites, it represents a significant body or work and set of tools and genomic resources that has the potential to accelerate discovery of antimalarial drugs that target the hypnozoite. *P. cynomolgi* is an excellent model system for the study of the *Plasmodium vivax* hypnozoite because of the degree of biological and genomic similarity between the species.

1) The authors mention that their work represents a second such analysis of the hypnozoite transcriptome, since the first analysis performed using laser capture and using the same strain of Pcynomolgi was published by Cubi et al. earlier this year. A direct comparison of the 120 transcripts described in that study with the transcripts identified in this study as a supplementary table in the Results section of the paper, and not just as a paragraph in the Discussion, would be very useful and contextualize the findings.

2) How can the authors distinguish between Pcyno-infected liver cells that never matured into schizonts and/or dying trophozoites, and hypnozoites? Because such parasite stages could complicate the differences seen in the transcripts between hypnozoites and schizonts. Is there a control analysis that could be used for this?

3) Some description of the success rate of the various infections would be really informative. For example, how many Pcyno infections of primary simian hepatocytes were done and how many successful? What was the approximate yield of Sz and Hz after cell sorting? Also, for each biological replicate, what was the concordance in identified transcripts, i.e. how much overlap between the replicates?

---

## [Author Response]

[Editors’ note: the author responses to the first round of peer review follow.]

Reviewer #1:Several comprehensive transcriptomic analyses have been performed and reported on all malaria stages, however this has been more problematic for the hardly accessible hypnozoite stage. One study has recently been published that reports the first transcriptome of hypnozoites and schizonts from infected monkey hepatocytes and optimized RNA‐seq analysis. The authors applied a laser dissection microscopy protocol to isolate individual Plasmodium cynomolgi hypnozoites Cubi R. et al., Cell Microbiol. 2017 Aug;19(8). Voorberg-van der Wel et al., work does not significantly depart from Cubi R et al., work but yet fills an important gap of knowledge on parasites dormancy. The problem of relapses due to dormant stages of Plasmodium vivax is among the most relevant aspects of malaria infection, which compromises all the efforts for eradication. Voorberg-van der Wel et al., provide a comprehensive transcriptomics resource of dormant and replicating malaria liver parasites that will be instrumental to the identification of drug targets for hypnozoites. The datasets generated are cautiously curated and analyzed with a clear explanation of the methods used.The authors report a considerably reduced use of 34% of Plasmodium physiological pathways, (91% are expressed in replicating schizonts). Importantly mitochondrial respiration for ATP homeostasis and maintenance of genome integrity are kept active. RNA fluorescence in situ hybridization allows a robust comparison of gene expression in schizonts and hypnozoites.Overall the results are more stringently interpreted than in Cubi R. et al. with only a dozen genes enhanced in quiescent hypnozoites while the expression of thousands of genes is repressed compared the liver schizonts. The authors identified two protein markers that differentiate quiescent from actively dividing parasites but actually failed to identify a hypnozoite specific marker. For example GAP45 is expressed in all invasive and motile stages of the malaria parasites include late liver stage schizonts.This work is not entirely novel and preliminary but should be of interest to readers of eLife who are interested in malaria and infectious diseases in general.

We thank the reviewer for this thorough review. We have now included a comparative analysis of our data with the dataset reported by Cubi et al. establishing more clearly that our contribution is a significant advance from their preliminary report. See subsection “Hypnozoites express a smaller set of genes than schizonts”, second paragraph; Discussion, second paragraph; subsection “Comparison with published data”, and in Figure 1—figure supplement 1.

1) The use of primaquine or tafenoquine as perturbators of the transcriptome would be a relevant and informative extension of this work.

We thank the reviewer for the suggestion to look at the transcriptional response to 8-aminoquinolines treatment. These experiments present a number of methodological challenges, one of which being that drug treatment is likely to result in a rapid developmental arrest that will reduce the number of cells available at day 7. As a result it might be difficult to extract interpretable RNAseq data that could be compared to the available dataset. Nonetheless we believe it is a good suggestion and this could be the scope of additional follow-up studies.

2) The data on copper chelator neocuproine was cidal to all liver stage parasites is preliminary and would deserve more robust investigation towards eradication of hypnozoites.

To further substantiate the neocuproine data we have now included the data of a third separate neocuproine drug test (Figure 4). We did these drug tests with the copper chelator neocuproine, because the high FPKM values in the RNAseq dataset for two putative copper transporters in both hypnozoites and schizonts suggested a role for copper in liver stage biology. Indeed the results show cidal activity of neocuproine in line with an important role for copper homeostasis in the liver stages. We included these data as a chemical validation step of the RNAseq data, which illustrates how our RNAseq data may point to biological processes important for hypnozoites. We feel that further validation of this compound as a drug for the elimination of hypnozoites is out of scope of this paper. We agree with the reviewer that indeed that would require much more drug testing at different time points and careful testing of toxicity both in vitro an in vivo.

Figure 4 is updated; legend amended; Text in manuscript updated (subsection “Expression pattern of potential drug targets in *P. cynomolgi* liver stages”, last paragraph).

Reviewer #2:The manuscript describes the transcriptomes of P. cynomolgi liver stages grown in primary Simian hepatocytes. This study is of interest because relapsing malaria parasites, such as P. cynomolgi, form a small, non-replicating dormant form of the parasite in the liver called hypnozoite that can reactivate and initiate relapsing malaria infection. The authors report the transcriptomes of replicating schizonts and persisting hypnozoites, which they are able to differentially separate using FACS sorting and a transgenic parasite strain expressing GFP. They identify several pathways expressed in the hypnozoite and validate expression of a few genes both by RNA FISH and translation at the protein level using antibodies. Interestingly, two copper transporters were robustly expressed in the hypnozoite as well as schizont. Treating cultures with a copper chelator reduced the number of hypnozoites and schizonts in a dose-dependent manner when dosed prophylactically. While the work presented is important to the malaria field, the authors do not identify any transcripts specific to the hypnozoite. Furthermore, a P. cynomolgi hypnozoite transcriptome has already been published recently (Cubi et al. 2017). Therefore, the work might be suitable for a more specialized journal.

We thank the reviewer and we believe that we have now addressed clearly in the manuscript the main objections raised by the reviewer and more specifically provide an extensive comparative analysis with the Cubi et al. previous analysis that clearly establishes the significant contributions provided in the comparative transcriptomics dataset we report here. See subsection “Hypnozoites express a smaller set of genes than schizonts”, second paragraph; Discussion, second paragraph; subsection “Comparison with published data”, and in Figure 1—figure supplement 1.

1) To identify differentially expressed genes in P. cynomolgi schizonts compared to hypnozoites the authors normalized the RNAseq FPKM's to the amount of host RNA present in the host hepatocyte because schizonts have many more genome copies and as such many more transcriptionally active units than hypnozoites. There are three major concerns with this method. 1) Schizonts will all have differing numbers of transcriptionally active units dependent on maturation stage. 2) It is expected that dramatic changes will occur between day 6 and 7 timepoints of analysis) during the schizont life cycle as the parasite prepares for segmentation and cytokinesis. 3) Most importantly, the quantity of host hepatocyte RNA is expected to be dramatically different in hypnozoite-infected hepatocyte versus a massive schizont-infected hepatocyte, with the latter containing much less host RNA. In consequence, any claims about transcriptional repression in hypnozoites cannot be supported by the data and should be removed from the manuscript or a convincing analytical strategy should be presented that fully supports the conclusions.

The reviewer expressed some legitimate concerns regarding the robustness of our claim of an intrinsically lower level of transcriptional activity for the hypnozoite. We recognize that there are confounders that substantially complicate the differential comparison of the transcriptome of the liver stages (e.g. higher number of transcriptional units in the schizont, heterogeneity of schizonts population in terms of number of nuclei). For that reason, we have indeed refrained to perform an extensive analysis of the relative changes in expression between schizont and hypnozoite and instead chose to describe the data as atlas of the genes expressed (or not) in schizonts and hypnozoites, rather than inferring functional role for genes based on their differential expression in liver-stages.

We found that 78.4% of the total RNA-seq reads of schizont-infected hepatocytes map to the host genome while 98.3% do so for the hypnozoite-infected hepatocytes. Because it is not possible to determine the actual number of active transcriptional units in the schizont and in order to control for the lower RNA content in the hypnozoite which could possibly be attributed to its lower number of transcriptional units, we decided to take a very conservative normalization approach and normalize the data against the host RNA content (a similar approach has been proposed by Westermann et al., PLoS Pathog. 2017). This approach is based on two main assumptions, first we assume that we capture the RNA content of a single host cell with each FACS event and second that the total RNA content of a host cell does not differ significantly between cells that are infected either with a schizont or a hypnozoite.

The reviewer states that the schizont-infected hepatocytes express much less host RNA. We are aware of relative changes in host cell gene expression upon infection and as the parasite matures in the hepatocytes as reported in several studies (PMC3619000, PMC4096771 and PMID:17981117), but to our knowledge there are no report of changes in total RNA content in response to the different developmental stages of the parasite in the hepatocyte. After normalization we found the average gene expression in the hypnozoite to be lower than in the schizont and we were able to confirm the general directionality of these observations using RNA-FISH.

To recognize the caveats highlighted by this reviewer, we have acknowledged all of those throughout the revised manuscript and soften the statement we made regarding the lower level of transcription observed in the hypnozoites.

2) The authors next conduct FISH to support their differential transcript abundance analysis conclusions drawn from RNAseq data. There are two issues with the FISH conclusions. First, FISH is not a quantitative method and in consequence cannot validate quantitative conclusions from RNAseq. Second, even if FISH would allow quantitative analysis, the schizont versus hypnozoite comparison is problematic in that the former contains thousands of transcriptionally active genome copies whilst the latter contains one active genome copy.

The reviewer suggests that FISH is not a quantitative method. We would like to point out that there are many methods for performing FISH, each with different characteristics (reviewed in Gaspar I, Ephrussi A. Strength in numbers: quantitative single‐molecule RNA detection assays. Wiley Interdisciplinary Reviews Developmental Biology. 2015;4(2):135-150.)

We have chosen an RNA-FISH technology, RNAscope from Advanced Cell Diagnostics, which has been validated as a quantitative method by Battich et al., who has performed RNAscope FISH analysis of more than 900 different human transcripts (Battich N et al. Image-based transcriptomics in thousands of single human cells at single-molecule resolution. Nat. Methods 2013 Nov; 10 (11):112733.). Battich et al. compared the quantities of most transcripts detected by RNAscope and showed excellent correlation with those obtained by RNAseq.

As stated above we have taken a conservative approach and chosen to perform a group-wise normalization procedure for the RNAseq data and the FPKM values after normalization procedure seem to generally agree with our FISH data. Indeed, the observed differences in RNA levels between schizonts and hypnozoites may be due to differences in cell size, differences in transcriptional units and/or by repression of transcription in (dormant) hypnozoites. As we cannot distinguish between these, we have now changed the wording in the revised manuscript to highlight those possibilities.

3) Do the genes listed in Supplementary file 5 meet any criteria for being differentially expressed? For example, an FDR or FC cutoff? If not, this analysis should be included.

We thank the reviewer for the questions as they highlighted the need for us to be more specific on the content of Supplementary file 5. The previous version of this table presented the results of the differential analysis of all *P. cynomolgi* genes regardless of whether their changes of expression were significant or not. In another words, all genes were listed in the table also those which did not meet the criteria for being differentially expressed. Now, we modified Supplementary file 5 to list only genes that meet the criteria to be differentially expressed between hypnozoites and liver schizonts using a cut-off of >2 fold change absolute value and a 10% false discovery rate.

See revised Supplementary file 5.

Do the authors expect the reported number of genes to be expressed in the liver stages of the parasite and are the transcripts not identified in either liver form expressed in blood stage schizonts?

To answer the second question: since these genes are found to be differentially expressed between hypnozoites and liver schizonts, they ought to be expressed in at least one liver stage (if not both). For completeness, along with the fold change and the p-value, for each differentially expressed gene we now report the average expression levels (e.g. avg FPKMs) in hypnozoites and liver schizonts with the scope to clarify whether the genes are expressed in one or both liver stages. Unfortunately we cannot report the expression levels of the same genes in the blood stage schizonts as we have not sequenced these samples.

4) For the Plasmodium pathway expression analysis, were any of the pathways determined to be significantly expressed? That is, are the pathways identified in the hypnozoite significantly expressed in hypnozoites compared to other pathways and are those pathways differentially present compared to schizonts? The statistical significance of the enriched pathways should be determined and reported.

To answer these questions, we performed overrepresentation analyses (enrichment via a hypergeometric test) on the genes that were highly expressed in hypnozoite and highly expressed in the schizont using annotation available in PlasmoDB and Gene Ontology (please see subsection “Pathway and Gene Ontology enrichment analyses” in Materials and methods and the first paragraph of the subsection “Hypnozoites express few core pathways including the physiological hallmarks of dormancy” in Results). The analyses revealed that while pathways related to RNA processing and ribosomal activity were enriched in both stages, pathways related to metabolic activity were only expressed in the schizont (*p* < 0.05 for all observations, *p*-values adjusted according to Benjamini-Hochberg procedure).

5) Much of the authors’ results are contradictory to a recently published P. cynomolgi liver stage transcriptome study (Cubi et al., 2017). It is important to give an honest assessment regarding the root causes for these discrepancies. Also move the comparative analysis into the Results section and discuss in the Discussion section.

Reviewers 2 and 3 raised criticisms pointing to the need for a more direct comparison of our data with the data reported by Cubi et al., we have compared the two data sets and found that there are substantial methodological and quality control issues with the Cubi et al. data. In addition, the direct comparison demonstrates the superior quality of the gene expression survey we report in our manuscript. See subsection “Hypnozoites express a smaller set of genes than schizonts”, second paragraph; Discussion, second paragraph; subsection “Comparison with published data”, and in Figure 1—figure supplement 1.

Reviewer #3:[…] 1) The authors mention that their work represents a second such analysis of the hypnozoite transcriptome, since the first analysis performed using laser capture and using the same strain of Pcynomolgi was published by Cubi et al. earlier this year. A direct comparison of the 120 transcripts described in that study with the transcripts identified in this study as a supplementary table in the Results section of the paper, and not just as a paragraph in the Discussion, would be very useful and contextualize the findings.

We thank the reviewer for the review and will most definitely make all the data publically available through NCBI Short Read Archive with the accession number indicated as soon as the paper is accepted for publication.

We have carried out an extensive comparison of our dataset with that of Cubi et al. We found the quality of the previously reported data to be of lower quality as it is now shown in Figure 1—figure supplement 1. See subsection “Hypnozoites express a smaller set of genes than schizonts”, second paragraph; Discussion, second paragraph; subsection “Comparison with published data”, and in Figure 1—figure supplement 1.

2) How can the authors distinguish between Pcyno-infected liver cells that never matured into schizonts and/or dying trophozoites, and hypnozoites? Because such parasite stages could complicate the differences seen in the transcripts between hypnozoites and schizonts. Is there a control analysis that could be used for this?

We acknowledge the points raised by the reviewer. At this juncture we cannot rule out the possibility that a (minor) part of the GFPlow sample contains small parasites that are dying/have never matured into schizonts. One could in theory treat the parasites with atovaquone and then FACsort. This would eliminate all parasites but the hypnozoites, however the drug treatment may also have an effect on the transcriptome. These experiments would be quite complex and substantially reduce the number of total parasites accessible.

From previous atovaquone treatments, we do already know that the majority of the small parasites indeed are hypnozoites (Zeeman et al., Antimicrob Agents Chemother. 2014;58(3):1586-95; Dembele et al., PLoS One. 2011 Mar 31;6(3):e18162). To further strengthen our data, we have performed a range of validation steps on several selected genes from the RNAseq dataset, both at the protein level and the RNA level. If GFPlow transcripts reflected the presence of nonhypnozoite populations, most likely this would have resulted in differential expression patterns in the small forms that are present in the cultures. During this validation process we have not come across any aberrant RNAseq data that could have resulted from the presence of non-hypnozoite forms.

Nonetheless, we do agree with the reviewer that this is an aspect of the analysis that should not be overlooked and this is why attempted to validate the RNAseq data set with orthogonal methods (e.g. IFA and RNA-FISH).

3) Some description of the success rate of the various infections would be really informative. For example, how many Pcyno infections of primary simian hepatocytes were done and how many successful? What was the approximate yield of Sz and Hz after cell sorting? Also, for each biological replicate, what was the concordance in identified transcripts, i.e. how much overlap between the replicates?

For the series of experiments relating to this paper we performed six blood stage infections. In two out of six blood stage infections, parasitemia was low (<0.2%) at the time of mosquito feeding. This resulted in poor sporozoite yields and not enough liver stage forms for FACSsort. The four other infections all resulted in successful liver stage infections with sufficient parasites for FACSsort. One of the infections was used for validation experiments. The events (yield of Sz and Hz) after cell sorting are presented in the existing Supplementary file 1 and the results of the alignment statistics are shown in the existing Supplementary file 3. We have now included this text in the Materials and methods section (subsection “Flow cytometry and cell sorting”).

The concordance in identified transcripts in the biological replicates was good, with only <16% of transcripts not shared between any of the Hz samples, and for Sz the concordance was even better as is shown in the Venn diagrams that we have now added (Figure 1—figure supplement 1). The concordance between the biological replicates is also shown by the correlation scores, ranging for Hz samples from 0.65 – 0.73 and for Sz samples from 0.86 – 0.92 as shown in the newly included scatter plots of the pairwise comparison (Figure 1—figure supplement 1).